# Gaussian Process Attention with Kernel Modeling

## Abstract

Transformers dominate NLP, yet their core component, self-attention, remains a heuristic, lacking a robust theoretical foundation. This paper reinterprets self-attention combined with rotary positional embeddings (RoPE) as an instance of Nadaraya-Watson (NW) kernel regression, unlocking a novel framework for enhancing attention through kernel modeling. We introduce Gaussian Process Attention (GPA), a novel technique, which leverages kernel theory to augment RoPE with a bank of decaying periodic functions. We demonstrate the result better captures the fine-grain characteristics of mutual information between tokens as a function of the distance between them. Tested on a GPT model using character-based tokenization and trained on a corpus of 14-million-characters, GPA outperforms baseline RoPE, reducing mean cross-entropy loss. More importantly, with only a nominal increase in the number of model parameters, GPA reduces computation time by 75% for an equivalent level of mean cross entropy loss. Moreover, we show that our GPA model opens a new avenue for studying mechanistic interpretability, revealing structures, such as paragraph lengths, and identifying redundant attention heads for model pruning. Our work bridges the application of kernel methods and the study of Transformers, providing a theoretical lens for analyzing self-attention while also delivering practical, scalable gains in performance and interpretability.

## 1 Introduction

The Transformer model Vaswani et al. (2017) has revolutionized artificial intelligence, and has become a key foundational architecture across diverse domains such as NLP Kalyan et al. (2021), computer vision Khan et al. (2022); Han et al. (2022), speech recognition Gulati et al. (2020), computational biology Zhang et al. (2023), and more. Nevertheless, Transformers remain more of a heuristic than a formal scientific framework. An underlying theory explaining not just how, but why they work has remained elusive, but such a theory is, arguably, essential for predicting safety, reliability, and alignment Bereska & Gavves (2024). Theoretical models are useful at several levels. They provide a basis for intuition, but more importantly, they establish a framework for analyzing errors and are a springboard for the innovation of new algorithms. The objective of this work is to present a modified self-attention model that both improves performance, decreases computation, and provides a set of new elements that can be used for interpretation of results in inference.

## 2 Methodology

### 2.1 Theory

The methodology in this work builds on Nadaraya-Watson (NW) regression Nadaraya (1964); Watson (1964), which uses a set of observed points, $\{x_i, y_i\}_{i=1}^N$, and a kernel function, $K_h$, to estimate the value of $y$ for any new point $x$. The estimate, $\hat{y}$, is computed as a normalized, weighted, shifted sum of the kernel function:

$$\hat{y} = \text{NW}(x) = \sum_{i=1}^{N} \left[ \frac{K_h(x - x_i)}{\sum_{j=1}^{N} K_h(x - x_j)} \right] y_i \qquad (1)$$

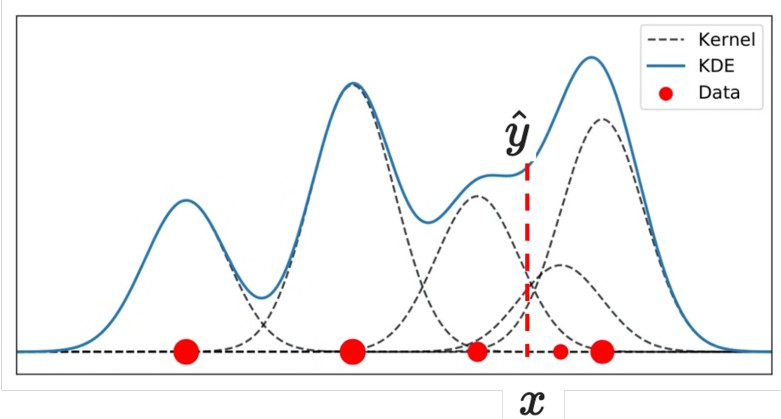

Figure 1: Illustrating Nadaraya-Watson regression. The resulting regression function, *KDE*, shown as a blue curve, is the weighted sum of shifted kernel functions, each shown as black, dashed curves. The data locations, $x_i$, are represented by the red dots on the horizontal axis, and their values, $y_i$ are represented by the size of the dots. The regression for a new data point, $x$, is shown on the regression curve as $\hat{y}$.

In this expression, the function, $K_h$, is centered around each of the $x_i$ and weighted by the corresponding $y_i$. This shifted and normalized weighted sum forms the regression function. The shape of $K_h$ is typically a symmetric, Gaussian-like curve whose width is controlled by a parameter $h$. In the context of Gaussian processes, the kernel function is equivalent to a wide-sense stationary covariance function with compact support Wong et al. (1988). Figure 1 illustrates a simple 1D example. When self-attention is implemented using rotary positional embeddings (RoPE) Su et al. (2024), its form is the same as NW regression:

$$\text{Att}(x_N) = \sum_{i=1}^{N} \left[ \frac{\exp(\frac{x_N^T Q^T \Theta^T \Theta K x_i}{\sqrt{d}})}{\sum_{j=1}^{N} \exp(\frac{x_N^T Q^T \Theta^T \Theta K x_j}{\sqrt{d}})} \right] V x_i \tag{2}$$

Here the $\{x_i\}_{i=1}^{N-1} \in \mathcal{R}^d$ are the context vectors and $x_N \in \mathcal{R}^d$ is the target vector. The matrices $Q$, $K$, and $V$ are learned parameters, and RoPE is a sparse matrix, $\Theta$, of fixed parameters that operates on the projected query, $q_n = Q x_n$, and key, $k_i = K x_i$ vectors. In Equation 2, RoPE is implemented by the matrix product $\Theta^T \Theta$, where $\Theta$ is block diagonal, where each block a $2 \times 2$ rotation matrix. The angles of rotation increase as a function of index and position. One of the key characteristics of RoPE is that $\Theta^T \Theta$ encodes relative positional information such that the attention score is modulated by the relative distance between sequence positions. The rotation applied to query and key vectors introduces a phase difference depending on their absolute positions, making the resulting dot-product attention scores sensitive to the positional difference between tokens in a way that typically causes attention to decay with distanceSu et al. (2024). Attention, and NW regression, both form normalized weighted sums dependent on relative distances. Thus, attention can be interpreted as a proper kernel function centered around each $x_i$ Tsai et al. (2019). Although attention is not symmetric, asymmetric kernels have been formalized in both theoretical frameworks and practical applications, and are useful for modeling conditional probabilities and directed graphs He et al. (2023b;a); Wu et al. (2010).

## 2.2 KERNEL MODELING

A useful characteristic of kernel functions is that they can be combined through summation or multiplication, and the result remains a valid kernel Aronszajn (1950). This is practical for modeling. RoPE is thought to implicitly embody the decaying periodic correlations known to be part of the structure of language Barbero et al. (2024), but it is a static structure. The goal of this section is to redesign attention by enriching RoPE with learned kernels to more flexibly model the mutual information between tokens. We begin by defining two kernel functions. The first, $P_k$, models

periodicity, and the second, $D_k$, exponential decay:

$$P_k(x_n, x_i) = \exp\left\{-2\alpha_k^2 \sin^2\left(\frac{|n-i|}{\tau_k}\right)\right\} \tag{3}$$

$$D_k(x_n, x_i) = \sigma_k^2 \exp\left\{-\frac{|n-i|}{l_k}\right\} \tag{4}$$

Each kernel is an explicit function of the indicial distance between the target and context vectors, $x_n$ and $x_i$, respectively. $P_k$ is a function of two learnable parameters, $\alpha_k$ and $\tau_k$, where the former controls amplitude and the latter period. $D_k$ depends on the learnable parameters, $\sigma_k$ and $l_k$, where the former is the strength of the term and the latter is a time constant or decay width parameter. The two kernels can be multiplied to model decaying periodicity, and a complex kernel function can be formed as the sum over a bank of $K$ such kernels:

$$G(x_n, x_i) = \sum_{k=1}^{K} D_k(x_n, x_i) P_k(x_n, x_i) \tag{5}$$

Finally, the expression in Equation 5 can be combined with that for attention from Equation 2 yielding a new kernel function, $\text{GPA}(x_n)$, given by:

$$\text{GPA}(x_n) = \sum_{i=1}^{N} \text{softmax}\left[G(x_n, x_i) + \frac{x_n^T Q^T \Theta^T \Theta K x_i}{\sqrt{d}}\right] V x_i \tag{6}$$

Because kernel functions are often used to represent Gaussian stochastic processes Wilson & Adams (2013), we call this model Gaussian Process Attention (GPA). In comparison with standard attention, it introduces $4K$ additional learnable parameters per attention head, which, as will be seen in Section 3.1, is only a nominal increase.

### 2.3 EXPERIMENTAL SETUP

Our study utilizes a compact GPT Transformer architecture composed of four layers, each featuring four attention heads. The experimental dataset comprises the complete works of Charles Dickens, sourced from Project Gutenberg Dickens (2018), and employs a character-level tokenization method as described in Banar et al. (2020). This corpus includes approximately 14 million characters and a vocabulary size of 93 tokens. By adopting this approach, we simplify the language preprocessing typically involved in training Transformer-based language models. Additionally, this method avoids the need to replace infrequent tokens with placeholders such as <UNK>, resulting in a concise and well-defined vocabulary. Our model incorporates the standard components found in Transformer blocks, including layer normalization, linear projection layers, multilayer perceptrons, as well as embedding and unembedding layers.[1] For our experiments, we set the context window length to 256 tokens and use an embedding dimension of 512. Model performance is evaluated using the mean cross-entropy (MCE) loss on the validation set.

## 3 EXPERIMENTAL RESULTS

### 3.1 TRAINING PERFORMANCE

Three experiments were run to evaluate the kernel models of the previous sections, and the results are shown in Figure 2. Each curve in the figure represents the MCE loss during training as applied to the validation data. The experiments consist of 200k gradient update iterations, with a batch size of 256 (equivalently 4 epochs). The data split is 90% for training and 10% for validation. The baseline experiment, represented by the blue curve, is the MCE loss of the GPT architecture when using standard self-attention with the RoPE implementation, as specified in Equation 2. The green curve is the MCE loss for our GPA formulation as described by Equation 6, where the bank is constructed from $K = 64$ decaying periodic kernels. The orange curve is an additional experiment that implements the GPA kernel bank but does not use RoPE, nor any other positional information other than that provided by the formulation of the kernel bank. The first observation is that the

---

[1] See https://transformer-circuits.pub/2021/framework/

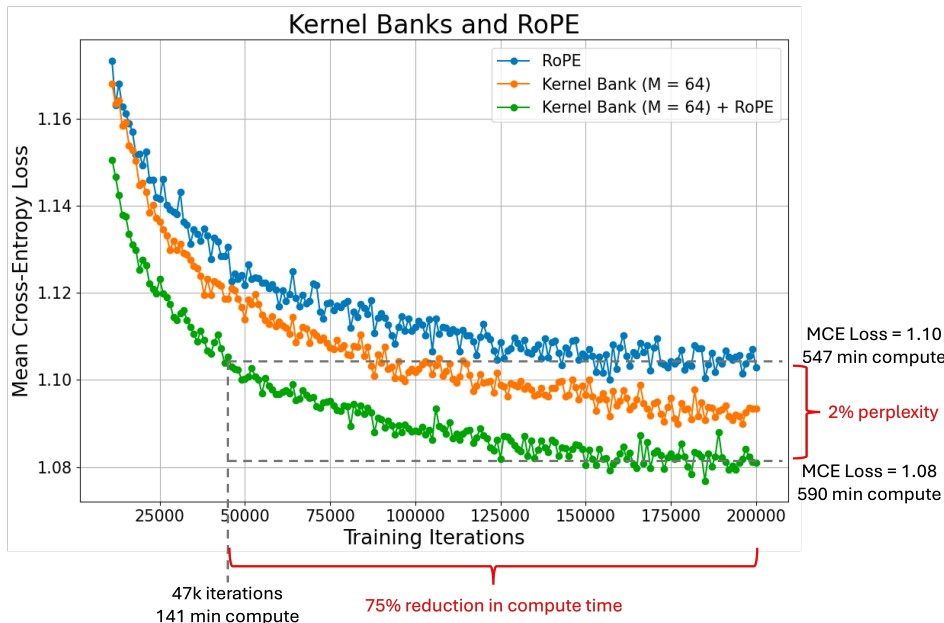

Figure 2: Comparison of MCE loss on validation data during training for the three experiments, each run for 200k iterations with a batch size of 256 (roughly 4 epochs). The blue curve represents the baseline implementation of self-attention combined with RoPE, the orange curve is for a GPA kernel bank excluding RoPE, and the green curve is the kernel bank combined with RoPE. There is a 2% improvement in perplexity between the baseline and the combined model, and the latter achieves the same degree of MCE loss in 75% less compute time.

kernel bank by itself outperforms RoPE for modeling relative positional information. It delivers a 1% improvement in perplexity at convergence. The best model, however, is the combination of the two methods, which achieves a 2% improvement in perplexity. This suggest the kernel bank and RoPE formulations are complementary models of data characteristics that neither is able to capture on its own. RoPE multiplicatively modulates attention scores based on the relative distance between tokens, and GPA additively shifts it. For this problem, the improvement in performance at convergence is modest, but more importantly, the combined method achieves the same level of MCE loss as the baseline in 75% less compute time. These improvements are annotated on Figure 2.

## 3.2 KERNEL BANK SHAPES

As detailed in Section 2.3, our model implements four layers with four attention heads per layer. For the standard self-attention model, the number of learnable parameters in our model works out to 12.7m weights. By comparison, in the GPA model, each of the 16 attention heads is modified by a kernel bank of 64 learned decaying periodic functions. This works out to $64 \times 4 \times 16 = 4096$ additional learned parameters, a nominal 0.03% increase in model size. Figure 3 shows the shape of the combined initialized kernel banks. The learnable parameters were initialized to $\alpha_k = 1$, $\sigma_k = 1$, and $l_k = 150$ for all $k$, and the $\tau_k$ take 64 evenly spaced values in the interval $[4, 192]$. As can be seen in the figure, the amplitude of the initial kernel bank falls about 50% (-3dB) at half the context window length. Figure 4 shows the functional form of each of the 16 kernel banks after 200k training iterations. The figure has a number of interesting features. The shapes of the four kernel banks in the first layer (shown in the upper left) are almost identical, and each has a prominent bump at lag 70. The second layer (upper right) is similar to the first, showing only small differences between the four heads. The third and fourth layers, however, more clearly differentiate the shapes between the heads. Notably, the strengths of the banks begin to vary, and in the fourth layer, one of the kernel banks is effectively zero. This suggests that the final layer of this model could discard one of the kernel banks without any loss of performance in inference.

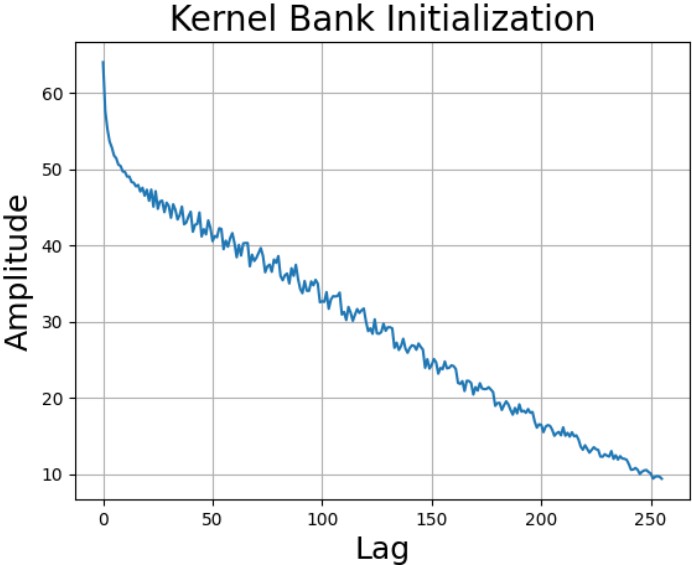

Figure 3: Functional shape of a kernel bank at initialization. Learnable parameters were initialized to make the overall amplitude of the kernel bank roughly 50% down at half the context window.

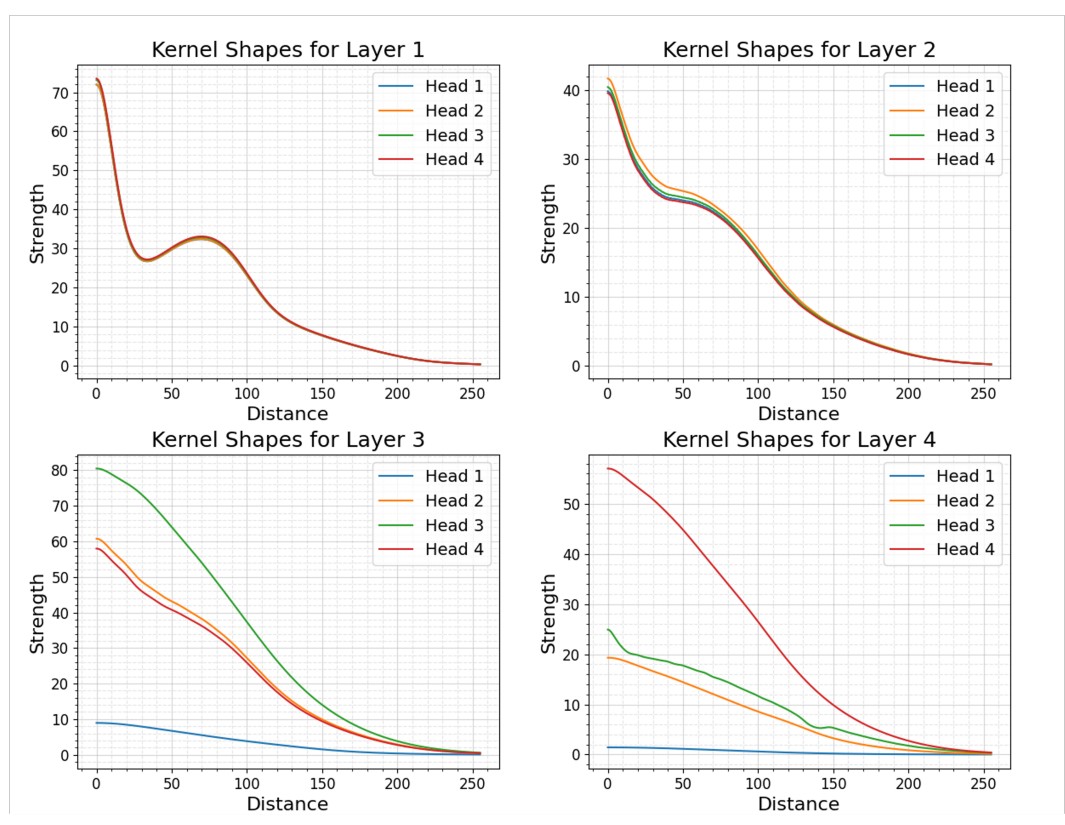

Figure 4: Shapes of the 16 trained kernel banks after 200k training iterations. The upper left corner shows the result for the four attention heads of the first layer, and the bottom right is the result for the fourth layer.

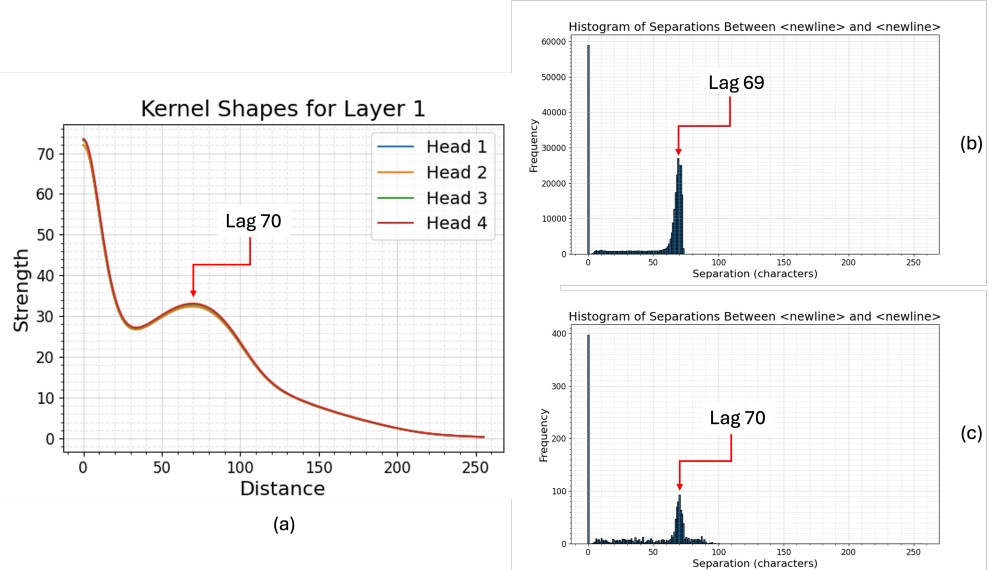

Figure 5: (a) Shapes of the trained kernel banks for the first layer of the model, showing a local peak at lag 70, and (b) The histogram of the separation between newline characters in the data, showing a peak at lag 69. Figure (c) shows the same histogram derived from data generated from the trained model, and its peak is exactly at lag 70.

### 3.3 Analyzing the Bump

As noted in the previous section, the kernel bank shapes for the first layer's attention heads have a prominent bump (see Figure 5(a)) with a peak value at lag 70. Our conjecture was that this bump corresponds to the mean paragraph length (in characters). To test this hypothesis, we computed the histogram of newline separations for the entire corpus. The result, shown in Figure 5(b), shows two spikes. The first is at a lag of one. This is due to the fact that paragraphs are separated by a double newline. The second spike occurs at lag 69. This supports the conjecture that the bump in the kernel shapes at lag 70 is related to the mean length of paragraphs. Figure 5(c) recomputes the histogram of separations between newlines computed from 50,000 characters generated by the trained model. This result puts the peak at precisely lag 70, further supporting the conjecture.

For the moment, it is not clear how such observed characteristics contribute to the interpretability of GPT models, but as reported by others, understanding how attention scores vary with distance between tokens helps reveal whether a model focuses more on nearby context or long-range dependencies, clarifying a model's processing strategy for language structure and semantics. For example, as reported in Hosseini et al. (2025), distance-based analysis can help identify which tokens the model deems important for predicting or contextualizing a target token, enabling token-level relevance or attribution methods to visualize model decision pathways. Moreover, tokens semantically or syntactically related often show particular distance patterns in attention. Understanding these can expose latent structures learned by the model and facilitate semantic interpretability.

### 4 Conclusions

In this paper, we demonstrate how self-attention combined with RoPE can be interpreted as an instance of Nadaraya-Watson regression, a modeling technique based on a sum of kernel functions. As positive-definite kernel functions can be interpreted as covariance processes, they are flexible modeling tools that can be combined using addition and multiplication. We leverage this characteristic to define a learnable kernel bank of decaying periodic functions and use this to better capture characteristics of natural language. When used within a GPT architecture for next token prediction, our experiments show that kernel functions are able to improve on the performance of RoPE and to model additional information in the data. The kernel bank improves performance, reducing

perplexity by 2% in our experiments, but more importantly, delivers an equivalent level of mean cross-entropy loss in 75% less compute time, demonstrating that the approach is a significantly better model of the data. The computational cost of this additional predictive power is nominal, augmenting the number of learned parameters by just 0.03%. In addition to improved performance, the kernel banks provide new opportunities for mechanistic interpretability. We studied a notable feature observed in a trained kernel function, and showed that it correlates to a specific feature in the data. Moreover, after training, we observed one of the kernel functions was uniformly zero, suggesting that its attention head was redundant. This is a valuable insight because it means that this head and its MLP can be removed for both training and inference, and in so doing, reduce associated computational and memory costs.

The results presented in this paper seem promising, but are for a small corpus and a small GPT model. Experiments with a larger corpus (for example, Wikipedia or Fineweb Penedo et al. (2024)) would validate the kernel bank efficacy for a more consequential dataset. The character-based tokenization strategy used for this paper was useful, as it allowed us to circumvent the many design and engineering questions related to vocabulary size that come with more sophisticated tokenization schemes such as WordPiece Schuster & Nakajima (2012) or byte-pair encoding Sennrich et al. (2016). The tradeoff is that character-based tokenization loses much of the semantic information derived from words. Finally, the kernel bank models need to be tested in downstream applications. Doing so would provide additional insight into their strengths, weaknesses, and capabilities.

## ACKNOWLEDGMENTS

Removed for anonymity.

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
