# OpenReview forum: "Gaussian Process Attention with Kernel Modeling"
_ICLR.cc/2026/Conference — Submitted to ICLR 2026_

### Official Review · Reviewer_hepz · 2025-10-17

**Soundness:** 2
**Presentation:** 2
**Contribution:** 1
**Rating:** 2
**Confidence:** 4

**Summary:**

Thanks for the submission. The paper considers position encodings based on modulating the attention logits between tokens i and n by a function of $|i-n|$, with this function parameterised as a sum of products of learnable ‘kernel banks’ (Eqs 3 and 4) – essentially, an extra relative position encoding (RPE). Compared to if RoPE is used on its own, this extra encoding improves performance in a small GPT model. The authors interpret the ‘bump’ in the learned RPE function as corresponding to mean paragraph length.

**Strengths:**

Training a small GPT model is a nice (and fairly ambitious!) experiment. Improving position encodings is an interesting and important research direction. Interpret learned RPE patterns and relating them to the underlying text structure makes good sense.

**Weaknesses:**

1. ‘Gaussian process attention’ is a misnomer. The paper is unrelated to GPs – stochastic processes for which all finite collections of random variables are jointly normally distributed. The authors use ‘GP’ because GPs use kernels and they parameterise their new position encoding using kernels (line 129). To me, this doesn’t make much sense, and is a little misleading. There's nothing probabilistic here.
2. $G(x_n, x_i)$ is a learnable function of $n - i$, parameterised using functions $P_k$ and $D_k$. As Eq 6 makes clear, this is incorporated as a relative position encoding (though I don’t think the authors ever explicitly describe it as such). RPEs are not novel.
If the contribution of the paper is actually a different parameterisation of RPEs, I think the text and experiments should be reworked to reflect this.
3. I also think the authors should compare to more sensible baselines, e.g. different existing RPE parameterisations. It’s not at all surprising that RoPE + RPE beats RoPE on its own – really, we want to know whether this RPE is better than previous RPEs.
4. Minor points:  issues with citation formatting – if the citation is not to be read as part of the flow of the text, I think it ought to be in brackets.

In light of missing baselines and a misleading framing (GPs aren't involved and this is really just and RPE), I don’t think this paper should be accepted to ICLR.

**Questions:**

1. Have I missed the point here, or is paper indeed about a (possibly novel) RPE parameterisation?

---

### Official Review · Reviewer_Go2L · 2025-10-31

**Soundness:** 2
**Presentation:** 1
**Contribution:** 1
**Rating:** 2
**Confidence:** 4

**Summary:**

The paper models attention as an instance of kernel density regression, which provides a framework to further analyze and interpret transformers. Built on this foundation, the paper proposes Gaussian Process Attention, a type of kernel attention combined with rotationary positional embedding (RoPE). Language modelling experiments on a text dataset show marginally better result than RoPE.

**Strengths:**

The paper provide some interesting empirical analyses on the influence of the text data distribution on the performance of the proposed method.

**Weaknesses:**

- Representing self-attention as nonparametric kernel density regression is not a novel idea. Several works in the literature have pointed out this interpretation of self-attention [1].
- The paper aims to interpret self-attention as kernel regression, however it lacks entirely the relevant discussion on transformer interpretability [1, 5] and Gaussian Process representation of attention [2, 3, 4].
- The paper suffers from sub-standard writing quality. The introduction is particularly poorly written with few convincing arguments to motivate the method.
- The experiments in the paper do not include relevant baselines such as kernel attention [5], Gaussian Process attention [2,3,4], making it difficult to evaluate the effectiveness of the proposed method.
- The experiments are very limited to just one text dataset. It is crucial to include more datasets in diverse settings to show the efficacy of the proposed method.
- I do not see the reason why the authors refer to this model as a Gaussian Process attention since it does not compute any uncertainty or covariance and the resulting attention (GPA in the paper) is essentially the mean function of a Gaussian Process.

[1] FourierFormer: Transformer Meets Generalized Fourier Integral Theorem
[2] Revisiting Kernel Attention with Correlated Gaussian Process Representation
[3] Calibrating Transformers via Sparse Gaussian Processes
[4] Self-Attention through Kernel-Eigen Pair Sparse Variational Gaussian Processes
[5] Transformer Dissection: A Unified Understanding of Transformer's Attention via the Lens of Kernel

**Questions:**

See Weaknesses

---

### Official Review · Reviewer_TBx3 · 2025-11-01

**Soundness:** 1
**Presentation:** 2
**Contribution:** 2
**Rating:** 2
**Confidence:** 3

**Summary:**

This work assesses the self attention in transformers as a form of kernel regression, and presents a novel augmentation called Gaussian Process Attention.  Results demonstrate improved accuracy and efficiency.

**Strengths:**

The formulation of rope based attention in the vein of Nadaraya-Watson kernel regression is an interesting and potentially highly impactful research direction.

The claimed 75% reduction in compute time is substantial, showing the potential to significantly improve convergence rates if this result can be generalised.

**Weaknesses:**

As emphasised in the Conclusion, the experiments are limited to a small corpus on a small model. A necessary first step, but the gulf between this character-level tokenisation approach and a modern LLM is so great that the claims outlined in the abstract feel very overstated.

The work is lacking in ablation studies. While it is interesting to speculate on the paragraph structure's impact on the kernel, what would be more compelling of course is to perform more experiments, e.g. retrain on a corpus with a different paragraph style and verify whether the bump moves as expected.

The introduction is incomplete and does not adequately cover the existing literature in this area - there have been very many attempts to connect attention and kernel methods, including NW regression. eg see "Elliptical Attention" (Nielsen et al., 2024); Correlated GP Transformer (Bui et al., 2024).

**Questions:**

What other kernel designs have you considered beyond the decaying periodic?

How do you see your work in relation to other recent works exploring the connection between kernels and attention?

---

### Official Review · Reviewer_r6Wh · 2025-11-05

**Soundness:** 3
**Presentation:** 3
**Contribution:** 3
**Rating:** 4
**Confidence:** 4

**Summary:**

This paper reinterprets self-attention with Rotary Positional Embeddings (RoPe) as a form of Nadaraya-Watson kernel regression. Based on this view, the authors propose Gaussian Process Attention (GPA), which augments RoPe with a learned bank of decaying periodic kernels designed to more flexibly model how mutual information between tokens varies with positional distance.
The kernel bank consists of K decaying periodic kernels, each parameterized by an amplitude, decay scale, and wavelength, enabling the model to learn richer patterns of long-range and periodic dependencies than RoPe alone. GPA modifies the attention score by additively combining the RoPE-modulated dot-product score with this kernel mixture. The model is evaluated on a small character-level GPT, for the task of next token prediction. The authors report: ~2% improvement in perplexity over RoPe alone and ~75% reduction in compute required to reach equivalent validation loss, with only 0.03% increase in parameter count. Additionally, the structure of the learned kernels reveals interpretable patterns (e.g. a peak at lag ≈70 corresponding to average paragraph length), and some heads effectively collapse, suggesting potential for attention-head pruning.

**Strengths:**

- The paper provides an intuitive reinterpretation of RoPe based attention as a form of Nadaraya–Watson kernel regression.
- The proposed GP Attn. only introduces around 4k additional parameters per headcmaking the method practical, lightweight, and easy to integrate into existing transformer architectures.
- The combined RoPe + kernel bank model reaches the same cross-entropy performance as the baseline with ~75% fewer training steps, demonstrating meaningful improvements in training compute efficiency.
- Analysis of the learned kernel shapes reveals alignment with structural properties of the data, e.g. the bump in the kernel shape at lag 70 corresponding to avg. new line separation length.

**Weaknesses:**

- While the bump at lag ~70 is interesting, I feel, the interpretability claims are anecdotal rather than systematically developed. Additional analysis would be needed to establish whether GPA consistently yields more interpretable heads or whether such patterns are dependent on the data.
- Not sure about the evaluation framework and character level tokenisation task that the authors present. There may be a reason that the authors pick this style rather than the traditional tokenisation in current GPT style next token prediction models. Would the results generalise to the traditional subword tokenisation setting?
- For a rigorous experimental suite there should be some mention of why there are no comparisons to other kernlised attention approaches or positional alternatives.

**Questions:**

- The reported ~75% reduction in compute arises from faster convergence, not from a reduction in FLOPs or memory per step?
- It is difficult to assess whether GPA is competitive relative to existing efficient attention methods...without any comparisons to baselines.
- The kernel bank uses K=64 decaying periodic kernels per attention head, but many of the learned kernels appear qualitatively similar (Figure 4). Could the authors provide an ablation over K? to understand if the performance benefits require a large kernel bank?

---

### Meta-Review · Area_Chair_gFtt · 2026-01-05

**Summary:**

This paper reinterprets self-attention with Rotary Positional Embeddings (RoPE) through the lens of Nadaraya–Watson kernel regression. Building on this perspective, the authors introduce Gaussian Process Attention (GPA), which enhances RoPE using a learned mixture of decaying periodic kernels that capture how mutual information between tokens varies with positional distance. Each of the kernels is parameterized by an amplitude, decay rate, and wavelength, allowing the model to represent richer long-range and periodic structures than RoPE alone. GPA adjusts the attention scores by adding this kernel mixture to the RoPE-modulated dot-product score.

While the formulation is intuitive, the core idea is not novel, as similar interpretations have been explored in prior work such as FourierFormer. The submission is also incomplete: at under seven pages, it falls well short of the ICLR page limit and contains formatting issues, citation errors, and an underdeveloped introduction, indicating the paper is not yet ready for review. The experimental evaluation is limited to small models and a single dataset, lacks ablations, and omits essential baselines from both kernel attention and Gaussian Process–based attention, making it difficult to assess the effectiveness of the proposed approach. Reviewers also question the terminology, noting that the method does not model GP covariance or uncertainty and essentially corresponds to a GP mean function, undermining the justification for calling it “Gaussian Process attention.” Three reviewers recommend rejection and one rates the paper marginally below the acceptance threshold. As the authors did not provide a rebuttal, none of these concerns were addressed. Given the limited novelty, insufficient experiments, incomplete presentation, and lack of response, I agree with the reviewers and recommend rejection.

**Reviewer Concerns:**

The authors did not provide a rebuttal, so none of the reviewers' concerns is addressed.

**Reviewer Scores:**

Since there is no rebuttal and discussion from the authors, I think the reviewers’ scores would have not changed even if a full discussion period had occurred.

---

### Decision · Program_Chairs · 2026-01-26

Reject